# Remaining in School in Rural China: Social Capital and Academic Self-Efficacy

Lucy P. Jordan [1], Xiaochen Zhou [1,*], Lue Fang [2], Qiaobing Wu [3] and Qiang Ren [4]

[1] Department of Social Work and Social Administration, University of Hong Kong,
Hong Kong SAR 999077, China; jordanlp@hku.hk

[2] Asia Research Institute, National University of Singapore, Singapore 119077, Singapore;
luf972@mail.harvard.edu

[3] Department of Applied Social Sciences, Hong Kong Polytechnic University, Hong Kong SAR 999077, China;
qiaobing.wu@polyu.edu.hk

[4] Center for Social Research, Guanghua School of Management, Peking University, Beijing 100871, China;
renqiang@pku.edu.cn

* Correspondence: xczhou@hku.hk

**Abstract:** Despite the significant economic and social transformation, students from rural China continue to remain at significant risk of early school leaving. Little research has addressed the jointly protective roles of family and school resources as well as children's sense of capability that may increase the likelihood of remaining in school. Data are drawn from the first two waves of a national probability sample of the China Family Panel Studies (CFPS) collected in 2010 and 2012, including households with youth aged 10–15 in 2010 living in rural areas (*n* = 1503). The results of logistic regression models predicting the likelihood of remaining in school during transition phases of secondary schooling highlight the importance of academic self-efficacy and social capital generated through youth peer networks. Given the severe negative consequences of not finishing secondary school for young people, especially during the later stage of economic transition unfolding in China, findings from this study contribute to an enhanced understanding of the "pull" factors that avert early school leaving. The study offers insight into the design of future low-cost targeted intervention strategies in rural China which could be applied to other middle-income countries.

**Keywords:** social capital; school dropout; rural China; academic self-efficacy

## 1. Introduction

Education is an important key to unlocking later human capital achievements including entry to and progression in the labor market [1]. The ramifications of school drop-out- are far-reaching at both individual and societal levels. At the beginning of the 21st century, the Chinese government took a series of countermeasures to reduce school drop-out. One of the most significant measures was the national policy "*liang mian yi bu*" (two waivers, one subsidy), whereby the central government budgeted to cover the expenditure of textbooks and miscellaneous fees for students from underdeveloped rural areas, with the budget rising from 200 million in 2002, 400 million in 2003, 1.17 billion in 2004, 7.2 billion in 2005 [2,3].

However, cumulative research suggests that students in rural China are dropping out of junior secondary school at troubling rates [4,5]. The official source suggested the junior high school dropout was 6.6% [6], while independent researchers estimated 17.6% to 31% of students dropped out before the completion of junior secondary school [3,7,8], not to mention the remarkable proportion of hidden dropout students who had the student status but absent from the classrooms [9]. Hence, it remains an open question in terms of why some students still opt for early school leaving despite the policy support, and indeed, why this may be increasing in the period following the injections of financing. The high

prevalence of junior high school dropouts warrants further investigation on who drops out, why they drop out, and importantly, how to prevent them from dropping out.

For children from disadvantaged families, leaving schools implies participation in income-generating activities or household chores that can constitute an increase in family income [4,5], therefore, poorer, rural, and male students are more likely to drop-out [3,7,8]. In the meantime, social capital has been identified as an important protective factor of children's risk behavior. Evidence suggests that social capital generated through different networks including families, schools, and peer groups may contribute to better academic performance and prevent early school leaving [10–12]. In addition to the resources that children can obtain from school and family, the self-efficacy of the children may also be associated with their willingness to remain in school [13].

By using the first two waves of a nationally representative data—China Family Panel Survey (CFPS) collected in 2010 and 2012, this study applies the concepts of social capital and academic self-efficacy to examine the extent to which social and interpersonal factors may serve as protective factors that avert dropping out. Findings may contribute to the design of future intervention strategies in rural China.

### 1.1. Family, School Social Capital, and Remaining in School

Social capital is broadly defined as material and immaterial resources received by individuals through other social agents [14]. Coleman [15] first introduced the conceptual framework of social capital as an important factor in children's educational development. Coleman defined social capital by its function, and it mainly refers to the resources embedded in social relationships [15]. Coleman examined social capital in three forms: obligations and expectations, information channels, and social norms [15]. Closure of social networks, for instance, the intergenerational closure within the family, or the closure of peer networks within the school, facilitates social capital in the settings [15]. According to Coleman, a higher level of family and school social capital are influential determinates of children's education [15]. Coleman also suggests that social capital serves as a mechanism to transmit family human capital from parents to children. Prior research in China [16] has confirmed the salience of the social capital for migrant adolescent populations, in particular highlighting the significance of family, school, and community social capital in explaining variation in psychosocial adjustment and self-assessed academic performance, there remains a gap in understanding the role of social capital amongst the general youth population, as well as in regards to whether social capital protects rural children from dropping out from schools.

Derived from the concept of social capital, family social capital depends "both on the physical presence of adults in the family and on the attention given by the adults to the child" [15]. It mainly focuses on the parent–child relationship [17], including the time and effort spent between the two actors. A higher family social capital refers to the stronger bonds between parents and children, especially the physical presence of parents, their attention on the child, and the deep involvement in parenting [18]. Coleman [15] explored the impact of family social capital on high school drop-out arguing that high school drop-out can be explained by levels of social capital produced by family social ties and network closure. Lower family social capital such as distant parent–child relationships may place children at risk for poor educational outcomes [19,20], while warmth, affection, commitment, and emotional support within the family may have a positive influence on the education outcomes of children [21]. Therefore, we hypothesize that:

**Hypothesis 1.** *A higher level of family social capital is positively associated with remaining in school.*

School social capital referring to school-based networks and friendships is found to be particularly helpful in enhancing the schooling outcomes [10,11,22]. Research suggests that students' rationale for secondary school drop-out is influenced by psychological stress due, in part, to poor teacher–student relationships, and peer victimization [5,23]. Peer

relationships and acceptance is associated with school adjustment of the adolescents [24], and children with worse peer relationship at school are more likely to drop out [25]. Beyond peer groups, strong student–teacher relationships are associated with higher student engagement and are an important predictor of academic performance and school completion [3,19,20]. Therefore, we hypothesize that:

**Hypothesis 2.** *A higher level of school social capital is positively associated with remaining in school.*

### 1.2. Academic Self-Efficacy and Remaining in School

Self-efficacy refers to an individual's perception of their capabilities [26], and within an academic context, self-efficacy is often referred to as academic self-efficacy, which defines an individual's ability to attain educational goals [27]. Research has consistently shown that academic self-efficacy is a significant determinant of academic achievement [28,29]. Increased academic self-efficacy as measured by mastery of academic material was found to minimize the risk of dropping out for adolescents [13]. Academic self-efficacy represents relatively future-oriented and malleable perspectives of the self and its potential [30]. Students' educational expectation is a critical manifestation of academic self-efficacy, and it plays a critical role in the process of educational attainment. One mixed-method study in China [5] finds that a passive learning attitude is more influential than disadvantaged family background in association with drop-out. Students with higher educational expectations would stay in school longer and would have better academic outcomes [31] and lower dropout rate [32,33]. In light of the test-oriented educational system in China that puts students under great pressure for academic competition [34], holding a high educational expectation is a source of psychological assets that motivate students to be intrinsically interested in learning and stay in school despite of adversity. In addition, students with higher academic self-efficacy would devote more to their schoolwork [35], while students with low academic self-efficacy are found to be lacking positive motivation, and they normally do not achieve or attempt to pursue higher school achievement [36,37]. Academic self-efficacy is also found to be related to sense of school belongingness, self-regulation, and intrinsic motivation in school [38–40]. Due to structure and resource inequality, rural Chinese students are likely to learn in large class sizes or in mixed grades classes with low teacher–student ratios [41], students' effort and self-control are important in directing their attention toward learning [42]. The behavioral engagement linked with the learning motivations is requisite for school success for all students; therefore, devoting more studying effort and obeying the rules could be particularly conducive to school completion for students. Therefore, we hypothesize that:

**Hypothesis 3.** *A higher level of academic self-efficacy is positively associated with remaining in school.*

### 1.3. The Present Study

The present study aims to understand the joint effects as well as the relative importance of social capital from family and school, and academic self-efficacy in influencing school completion among rural junior high school students in China. We view social capital as a set of resources accessible through social networks. Recognizing the multifaceted nature of social connections in social capital, one of our main interests is to explore how social capital can operate across different contexts of family and school social capital may be associated with school dropout in rural China. Moreover, our study also captures children's personal agency of academic self-efficacy to understand the extent to which efficacious beliefs and effort regulation influence school completion above and beyond the effect of social capital. The conceptual framework of the current study is presented in Figure 1.

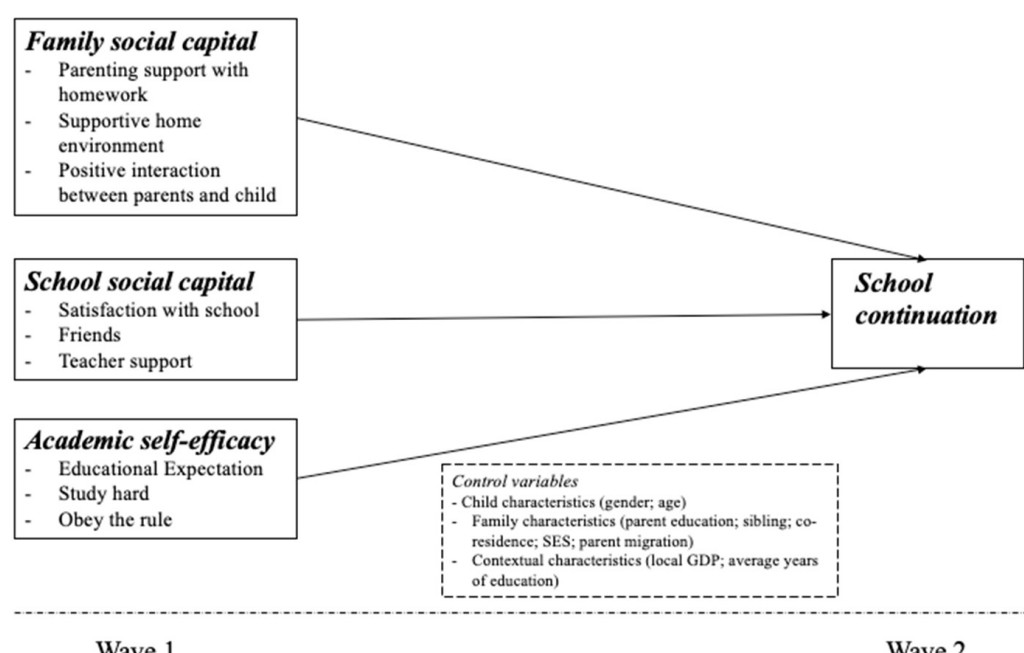

**Figure 1.** Conceptual framework of the study.

## 2. Materials and Methods

### 2.1. Data

This study draws two waves of a nationally representative dataset—China Family Panel Studies (CFPS) dataset collected in 2010 and 2012. Conducted by the Institute of Social Science Survey at Peking University (Retrieved 3 April 2020, from https://www.isss.pku.edu.cn/cfps/index.htm), the CFPS examines a range of topics relevant to the current study including educational participation, employment, family dynamics and relationships, and migration. The sampling design is a three-stage stratification based on county and community characteristics at stage 1 and stage 2, followed by mapping and selection of households at stage 3. All members over age 9 in a sampled household were interviewed. Five provinces (Gansu, Liaoning, Henan, Guangdong, and Shanghai) were selected for oversampling (1600 families in each) based on historical social and economic development and complemented with households from 20 additional provinces (including Jiangsu, Zhejiang, Fujian, Jiangxi, Anhui, Shandong, Hebei, Shanxi, Jilin, Heilongjiang, Guangxi Zhuang Autonomous Region, Hubei, Hunan, Sichuan, Guizhou, Yunnan, Tianjin, Beijing, Chong-qing, Shaanxi) to comprise a national population-based sample. In addition to community, household, and adult household surveys and importantly for this study, the CFPS has a self-report children module for children in sample households aged 10–15 (inclusive) years of age, and information from the children who answered this module in 2010 is the initial inclusion criteria for selecting the study sample analyzed here. This study only utilized Wave I and Wave II of CFPS because students in Wave III will be beyond high school graduation in Wave III.

### 2.2. Sample

The study sample for this research includes adolescents aged 10 to 15 at Wave 1 in 2010, inclusive, reflecting both primary school-age children approaching the first educational transition to lower secondary/middle school, and secondary school age adolescents at the completion of the official compulsory schooling in China of nine years. The sample was selected from the full household sample of the CFPS (*n* = 14,950) and based on the age at Wave 1 in 2010 (*n* = 1503).

*2.3. Measures*

Family social capital was drawn from a series of questions in within family parent–child interaction, including parenting behaviors and parental academic-related support. The first item includes parents' response to the question "how often do you supervise child homework at home." The second and third items involve interviewers' rating of supportive home environment (home environment and learning materials that indicates parents care about children's learning), and positive social interaction (whether parents take the initiatives to talk to the child).

School social capital was measured with three items: a binary indicator of whether child has any best friends; teacher support ("would you talk to your teachers when you have problems"); and self-reported overall satisfaction with school life ("are you satisfied with your school").

Academic self-efficacy was measured with three items. The educational expectation was recoded from the original eight possible response categories (primary-, middle-, secondary-school, 2/3-year college, 4-year university, master, doctor, and "no need to attend school") into four categories (primary-/middle-school/no need, secondary school, university degree or higher). Two additional items of child self-reported academic behavior (I study hard) and self-control (I obey the rules) were also included as measures of academic self-efficacy.

Demographic factors included child age, gender, hukou, family structure, household head's year of education, household income, and family structure. The educational levels of parents as well as number of siblings are also correlated with junior high school drop-out due to family resource allocation restrains suggesting the important role of social capital of the family [8]. The family income measure combines information from data across the 2010 CFPS and adopted from Xu et al. [43] incorporating different types of income including salary, business, asset, and transfers. Following Filmer and Pritchard [44] a three-category indicator of family income was generated, first by dividing income for the subsample into quintiles, and then combining the bottom two quintiles as well as the next middle two thus creating a 40-40-20 distribution to better reflect developing country economics. Three additional measures of family structure were created, a binary indicator of whether one or both parents are migrants, whether a grandparent lives in the household, and number of siblings to the index child. The following contextual characteristics are included as control variables in all the models to capture important sources of explanation at community level. From the 2011 Census at the country level, we include GDP per capita, average years of education, and we also used a measure of geographic location following the study design of the CFPS including Liaoning, Henan, Guangdong, Gansu, and other which includes the remaining 20 provinces [45].

*2.4. Analytical Approaches*

To obtain a general understanding of the broad characteristics of the data and to identify the dominant patterns in the associations among measures, we first conducted descriptive analyses. In the second stage, we used logistic regression of Wave 1 characteristics to predict the likelihood of children remaining in school at Wave 2 for the full children sample. Deviance statistics are used to compare the goodness-of-fit for nested models and alternative models that are not nested are then used to evaluate the final model [46], and lower AIC values indicate a better-fit model. Stata IC 12 (64-bit) is used to complete the analyses.

**3. Results**

Table 1 presents the overall sample characteristics of the current study by the in-school status of the children at Wave 2. It shows the overall sample description (column 2), the sample characteristics of children who remained in school at Wave 2 (column 3), and those who had left school at Wave 2 (column 4). It also presents the bivariate analysis results between the in-school and early leaver group in the last column, with the result significance

marked in stars. As shown in Table 1, the overall dropout rate between Wave 1 (2010) and Wave 2 (2012) was around 12%. Results from the t-test and chi-square test provided some insights into the differences in characteristics between those who drop out and those who remain in school. Across students of rural backgrounds, older and male students independently had a higher likelihood to leave school. Those who leave school are more likely to have lower family socioeconomic status (SES) indicated in family income and father's education. Having siblings was also related to a greater chance of leaving school. There was no significant difference regarding the contextual characteristics between the two groups.

**Table 1.** Sample characteristics by in-school status at Wave 2 (*n* = 1503).

| | Sample of Baseline | | In School at W2 | | Early Leaver at W2 | | |
|---|---|---|---|---|---|---|---|
| | M | SD | M | SD | M | SD | |
| Child characteristics | | | | | | | |
| Drop-out at W2 | 12% | - | | - | | - | |
| Age (12–17) | 14.5 | 1.7 | 14.4 | 1.7 | 15.5 | 1.5 | *** |
| Male | 49% | - | 48% | - | 58% | - | * |
| Family characteristics | | | | | | | |
| Father's education | 5.7 | 3.9 | 5.9 | 3.9 | 4.3 | 3.9 | *** |
| Family income (RMB) | 25,221 | 27,125 | 25,690 | 27,703 | 22,117 | 22,750 | * |
| Parent migrant | 6% | - | 6% | - | 5% | - | |
| Grandparent in household | 33% | - | 34% | - | 27% | - | |
| Number of siblings | 1.2 | 1.0 | 1.2 | 1.0 | 1.4 | 1.2 | ** |
| Contextual characteristics | | | | | | | |
| GDP per capita | 18,694.1 | 20,642.4 | 18,825.0 | 21,585.2 | 17,752.8 | 11,828.0 | |
| Average years education | 8.3 | 1.1 | 8.3 | 1.0 | 8.2 | 1.3 | |
| Province | | | | | | | |
| Liaoning | 2% | | 2% | | 3% | | |
| Henan | 9% | | 8% | | 11% | | |
| Guangdong | 8% | | 7% | | 9% | | |
| Gansu | 4% | | 4% | | 3% | | |
| Other # | 78% | | 78% | | 74% | | |

Notes: * *p* < 0.05, ** *p* < 0.01, *** *p* < 0.001 for *t*-tests and chi-square tests based on level of measurement; Data are from the China Family Panel Studies (CFPS), Wave 1 (2010) and Wave 2 (2012) (Retrieved 3 April 2020, from https://www.isss.pku.edu.cn/cfps/index.htm); W2 = Wave 2 (2012); M-mean; SD-standard deviation; Descriptive statistics are weighted; fourteen cases from Shanghai were dropped because of the high level of urban development in Shanghai municipality; # Other are from the additional 20 provinces listed in the data section.

*Multivariate Logistic Regression*

We aim to understand the independent and overall effects of social capital and academic self-efficacy on remaining in school. We therefore built a series of logistic regression models to better understand the unique contribution of blocks of measures. In the first block child characteristics, demographic factors were estimated (Table 2, Baseline Model A). Model B estimated the role of family social capital, and Model C examined the effects of school capital measures in association with remaining in school. The model D investigated the effects of academic self-efficacy measures of educational expectations, academic behavior, and self-control. The full model including all the blocks of variables discussed above is also presented in Table 2.

Of note is the stability of the baseline model characteristics with little change in the odds ratios, and consistency in statistical significance. As expected, there was an increased likelihood of dropping out as the children age, with the oldest children (age 16 and 17) more likely compared to the youngest (age 12). Child gender was also significant, with boys 68% more likely to leave school early, or 32% less likely to remain in school. Higher levels of father's education are associated with a greater likelihood of remaining in school,

whereas having siblings decreased the likelihood of remaining in school by 18%. Family income and the other measures of family structure were not significant.

**Table 2.** Predicting the odds of likelihood of staying in school (*n* = 1503).

| Variables | Full Model | | Model A Baseline Model | | Model B Family Capital | | Model C School Capital | | Model D Academic Self-Efficacy | |
|---|---|---|---|---|---|---|---|---|---|---|
| | OR | 95% CI | OR | 95% CI | OR | 95% CI | OR | 95% CI | OR | 95% CI |
| Child characteristics | | | | | | | | | | |
| Age_12 | | | | | | | | | | |
| 13 | 0.60 | 0.26, 1.35 | 0.70 | 0.32, 1.49 | 0.60 | 0.27, 1.34 | 0.58 | 0.26, 1.29 | 0.65 | 0.32, 1.54 |
| 14 | 0.37 * | 0.16, 0.83 | 0.43 * | 0.21, 0.88 | 0.39 * | 0.18, 0.86 | 0.38 * | 0.17, 0.83 | 0.39 * | 0.20, 0.91 |
| 15 | 0.21 *** | 0.10, 0.44 | 0.25 *** | 0.13, 0.48 | 0.22 *** | 0.11, 0.45 | 0.20 *** | 0.09, 0.41 | 0.25 *** | 0.13, 0.53 |
| 16 | 0.16 *** | 0.07, 0.33 | 0.16 *** | 0.08, 0.32 | 0.16 *** | 0.07, 0.33 | 0.14 *** | 0.06, 0.29 | 0.16 *** | 0.08, 0.33 |
| 17 | 0.11 *** | 0.05, 0.23 | 0.10 *** | 0.05, 0.19 | 0.11 *** | 0.06, 0.23 | 0.10 *** | 0.05, 0.20 | 0.11 *** | 0.05, 0.20 |
| Gender | 0.61 ** | 0.44, 0.86 | 0.68 ** | 0.50, 0.91 | 0.65 ** | 0.47, 0.89 | 0.62 ** | 0.45, 0.85 | 0.75 | 0.50, 0.93 |
| Family characteristics | | | | | | | | | | |
| Parent Education | 1.07 ** | 1.03, 1.12 | 1.11 *** | 1.07, 1.15 | 1.1 *** | 1.05, 1.14 | 1.10 *** | 1.06, 1.15 | 1.09 *** | 1.04, 1.12 |
| Number of siblings | 0.88 | 0.83, 1.66 | 0.82 ** | 0.72, 0.95 | 0.85 * | 0.74, 0.99 | 0.83 * | 0.72, 0.97 | 0.89 | 0.76, 1.02 |
| Co-residence | 1.18 | 0.75, 1.03 | 1.29 | 0.94, 1.77 | 1.14 | 0.81, 1.58 | 1.17 | 0.84, 1.63 | 1.25 | 0.76, 0.90 |
| Socioeconomic level | | | | | | | | | | |
| Medium | 0.95 | 0.66, 1.36 | 1.07 | 0.78, 1.47 | 0.97 | 0.68, 1.36 | 0.97 | 0.69, 1.36 | 1.07 | 0.75, 1.49 |
| High | 1.30 | 0.81, 2.09 | 1.29 | 0.85, 1.97 | 1.34 | 0.84, 2.12 | 1.32 | 0.84, 2.09 | 1.33 | 0.85, 2.07 |
| Parent Migration | 1.27 | 0.55, 2.89 | 1.09 | 0.52, 2.26 | 1.18 | 0.53, 2.62 | 1.07 | 0.50, 2.30 | 1.29 | 0.60, 2.79 |
| Family social capital | | | | | | | | | | |
| Parenting support with homework | 1.04 | 0.99, 1.09 | | | 1.05 * | 1.01, 1.11 | | | | |
| Supportive home environment | 1.08 | 0.83, 1.43 | | | 1.21 | 0.93, 1.59 | | | | |
| Positive interaction between parents and child | 0.89 | 0.67, 1.18 | | | 0.91 | 0.69, 1.20 | | | | |
| School social capital | | | | | | | | | | |
| Satisfaction with school | 0.89 | 0.76, 1.05 | | | | | 0.90 | 0.77, 1.05 | | |
| Friends | 1.74 * | 1.09, 2.79 | | | | | 1.60 * | 1.02, 2.52 | | |
| Teacher support | 0.69 | 0.24, 1.99 | | | | | 0.76 | 0.27, 2.13 | | |
| Academic self-efficacy | | | | | | | | | | |
| Educational Expectation (ref: primary or less) | | | | | | | | | | |
| Secondary | 1.47 *** | 1.27, 1.71 | | | | | | | 1.56 *** | 1.36, 1.80 |
| 2–3 year college | 1.10 *** | 0.59, 1.69 | | | | | | | 1.21 *** | 0.64, 2.13 |
| University or higher | 3.12 *** | 1.91, 5.10 | | | | | | | 3.34 *** | 1.96, 5.19 |
| Study hard | 1.06 | 0.98, 1.15 | | | | | | | 1.11 ** | 1.02, 1.19 |
| Obey the rule | 0.79 * | 0.64, 0.96 | | | | | | | 1.12 | 0.97, 1.30 |
| Constance | 9.85 | | 16.56 | | 13.58 | | 22.98 | | 1.87 | |
| PseudoR² | 0.14 | | 0.124 | | 0.108 | | 0.11 | | 0.185 | |
| AIC | 1058.891 | | 1226.99 | | 1104.39 | | 1113.809 | | 1129.611 | |

Note: * *p* < 0.05, ** *p* < 0.01, *** *p* < 0.001; Data are from the China Family Panel Studies (CFPS), Wave 1 (2010) and Wave 2 (2012) (Retrieved 3 April 2020, from https://www.isss.pku.edu.cn/cfps/index.htm).

Models B–D bring in the key substantive factors of family, school, and individual characteristics in relation to early school leaving. First, in Model B, parents' support with children's homework had a positive effect on children's staying in school; however, such effect became non-significant when considering other factors in the full model.

Second, Model C suggested the salience of supportive relationships, especially those with peers on remaining in school. Children who reported having best friends were more likely to remain in school. Relative to other predictors, the predicted increase in likelihood was quite strong with children who report close friendship relationships 1.60 times more likely to remain in school two years later. The magnitude of the predictive value of peer social capital became even stronger (1.74 times) in the full model highlighting the strong impact of peers for Chinese children above and beyond family and individual influences.

In Model D, child educational expectations, academic effort, and self-control emerge as important explanatory factors. Having a high educational expectation toward upper secondary school and higher education was strongly associated with the outcome: children who expected themselves to complete 4-year university degree or higher were more than three times as likely to remain in school. Self-reported academic effort had a positive association with staying in school in Model D, but the significance fell out of boundary when controlling for other family and school influences. Self-control was also found to be associated with an increased likelihood of remaining in school.

In the full model, the simultaneous multivariate logistic regression modeling was used to compare the differential independent influence of predictor variables on early school leaving. Consistent with the results from independent models discussed above, the full model denoted the protective role of pro-social friendship, high educational expectation, and positive self-control in nurturing resilience among Chinese children to remain in schooling within rural contexts.

## 4. Discussion

By applying information from two waves of a national dataset, the current study has attempted to shed some light on this important topic of early school living in the context of Chinese rural communities. Prior literature provided guidance about possible determinants of early school leaving, and summarized dropout as a negative outcome of intergenerational social disadvantage accumulation, which can have a long-term impact on children's transition to adulthood and their development over the life course [47]. The current study provides compelling evidence that various social capital from various social agents, especially peer support, and personal qualities related to academic self-efficacy are consequential to remaining in school for children in rural China. The implications are important, especially for consideration in economically disadvantaged settings, given the inter- and intra-personal nature of the key determinants.

### 4.1. Importance of Family and School Social Capital

In response to the H1, our findings suggest that family social capital solely has a significant association with remaining in school, that children whose parents support their homework were more likely to remain in school. As indicated in prior study, parental involvement might be one of the most significant factors to prevent early school leaving [48]. The mechanisms underneath could be, for example, that parents' academic involvement with homework is positively associated with Chinese students' academic performance and attitudes [49], which may further enhance children's academic engagement and motivate them to continue studying in school.

Our finding supports H2. The results revealed that school social capital generated from relationships with peers is an important determinant of remaining in school. This echoes prior studies that negative school experiences with peers may have a significant influence on children's decision of leaving school. For instance, qualitative evidence has indicated that without a close friend, children were more likely to feel ignored, invisible, isolated, and thus be less engaged in school and be more likely to drop out of school [25]. However, different from prior studies that highlighted the importance of teachers' role in preventing early school leaving, we did not find significant difference in terms of teachers' support and the satisfaction of school across the group who dropped out and remained in school. A tentative explanation might be that children in our sample are adolescents, and they started to establish their independence by switching their focus of social life from family to peer networks [50]. Peers become the center of children's life and might be more influential on children's choices.

### 4.2. Academic Self-Efficacy and Remaining in School

In addition to the important role of social resources gained from supportive peer and teacher relationships [51], this study also highlights the intra-personal factors of children

by exploring the role of academic self-efficacy. The finding supports the third hypothesis. Educational expectation is well recognized as an important measure linked with staying in school [52] and our study provides support that children with expectations of higher education are the most likely to remain in school. This measure, judged by impact on the change of odds ratio, is the strongest explanatory factor in the current study. Goals about achieving educational completion markers (graduate from secondary, university, higher) are an important link to understanding which children remain resilient to various environmental factors that could disrupt their educational trajectories.

Academic effort and obeying rules in school provide some speculative insight into this line of reasoning. Academic effort can be considered to have a direct effect on academic achievement as well as more indirect effects through, self-efficacy or self-concept [13]. Future research should expand the inquiry to include broader dimensions of self-efficacy, especially in the Chinese context, to determine how positive self-assessment could be nurtured and supported amongst children.

### 4.3. The Covariates and Remaining in School in Rural China

In this study, despite mean differences in wealth for households with children who remain in school compared to those who leave, controlling for other characteristics completely attenuates the bivariate difference. The finding is similar to prior work on drop out in upper-secondary technical and vocational education and training, which suggested that due to financial constraints, the poorest families may not be able to enter the educational program in the first place [53]. Furthermore, the insignificance of the wealth of the household does not necessarily imply that household socio-economic differences are not important, in fact, in our study, father's educational attainment illustrates the persistent significance of family SES. Children of the least educated fathers are most at risk for early school leaving. This is an important issue in a country, which has seen rising overall levels of education completion, including tertiary education, as these young people from rural families with the lowest levels of education likely represent a particularly "at-risk" group with the higher educational attainment within China being predominantly concentrated in more developed urbanized areas. A continuation of lower levels of educational completion for the younger generation is thus likely to perpetuate the intergenerational transmission of social inequality, rather than breaking the cycle and promoting social mobility.

### 4.4. Limitations and Implications

There are several limitations of the current study. While the use of secondary data allows for analysis of a national sample, the conceptual acuity is limited by the available measures. Future research should consider the inclusion of more nuanced measures. Another limitation of the data is related to the age of the respondents at the second wave. The primary source of data for the conceptual analysis is the self-report module asked only of children aged 10–15. The parents of children are only asked complementary questions about children's education for the same age range. This means that at Wave 2, children aged 16 and 17 do not have comparable information to those aged 12–15, thus restricting the available data for conceptually rich analyses. To address this limitation, our current study used information from Wave 1 to predict the likelihood of remaining in school two years later at Wave 2, making use of the detailed information from the first Wave as well as capitalizing on the inclusion of the older children experiencing the transition from lower to upper secondary school.

On balance, despite these limitations, the current study brings some potential implications. Practically, the study calls for parent(s) to maintain active connections with their children and improve their involvement in children's education. Second, it calls for school counseling and social work programs in rural China to enhance the school social capital, to promote children's school engagement and protect them from early school leaving. In addition, the current study also highlights the essential role of children's agency, which suggests more capacity building programs for children to enhance their academic self-efficacy

is necessary. For instance, goal setting, or peer mentoring programs could be potential low-cost interventions in rural schools to support bolstering students' self-determination of school completion. Last but not the least, beyond the discussion of dropping out of academic-track schools, promoting high-quality vocational education programs might also be helpful for the children who aim to enter the labor market via the technical track.

In conclusion, this study contributes to debates on the factors related to remaining in school among rural Chinese children coming of age in a period of unprecedented social and economic development. Future research can focus more deeply on the analysis of longitudinal factors as well as greater articulation of mechanisms operating within children's social capital. Better understandings of protective mechanisms that can prevent early school leaving can contribute in part to addressing growing social inequalities in education and subsequent labor market success and offer guidance in the development of social policy and practice planning to serve rural children in China and other developing contexts.

**Author Contributions:** Conceptualization, L.P.J., X.Z. and Q.R.; methodology, L.P.J., X.Z., L.F., Q.W. and Q.R.; formal analysis, L.P.J., L.F., X.Z. and Q.R.; writing—original draft preparation, L.P.J., L.F. and X.Z.; writing—review and editing, L.P.J., X.Z., L.F., Q.W. and Q.R.; funding acquisition, L.P.J. All authors have read and agreed to the published version of the manuscript.

**Funding:** This research received no specific grant from any funding agency in the public, commercial, or not-for-profit sectors.

**Institutional Review Board Statement:** This study adopts the dataset from China Family Panel Studies (CFPS), designed by Peking University (PU) research team, supported by PU 985 funds, and implemented by the Institute of Social Science Survey (ISSS) PU. No extra ethical approval was sought.

**Informed Consent Statement:** Informed consent was obtained from all subjects involved in the study.

**Data Availability Statement:** The data of China Family Panel Studies are available on: http://www.isss.pku.edu.cn/cfps/en/index.htm (accessed on 3 April 2020).

**Conflicts of Interest:** The authors declare that they have no conflict of interest.

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
