# Peer review of "Remaining in School in Rural China: Social Capital and Academic Self-Efficacy"

_2673-995X, doi:10.3390/youth2020011_

Round 1
Reviewer 1 Report
The article addresses a very interesting topic, applies an appropriate methodology and presents the results clearly. The purpose of the study is described in a logical, comprehensible, and explicit manner. The significance of the research is clearly established. Also, the authors presented the limitations of the research and highlighted the implications of this research.
However, the authors should also consider the following suggestions:
- Introduction
- I would suggest to the authors to write the research hypotheses (H1-H3) on a new line and to delimit them from the text of the paper in order to make it easier to visualize what is being tested through the paper.
- Results
- Line 191: because it is the first appearance in the text, the full name for the SES and the abbreviation in parentheses must be used. After that, the abbreviation can be written in the text of the paper.
- Line 195: Table 1 - The abbreviations used in Table 1 are not indicated. It is recommended to include a table foot with the meaning of M and SD.
- Also, for Table 1 and Table 2 the authors should mention the source.
Author Response
Response to Reviewer 1 Comments
The article addresses a very interesting topic, applies an appropriate methodology and presents the results clearly. The purpose of the study is described in a logical, comprehensible, and explicit manner. The significance of the research is clearly established. Also, the authors presented the limitations of the research and highlighted the implications of this research.
Response: Thank you so much for your comment.
However, the authors should also consider the following suggestions:
- Introduction
- I would suggest to the authors to write the research hypotheses (H1-H3) on a new line and to delimit them from the text of the paper in order to make it easier to visualize what is being tested through the paper.
Response: Thank you so much for your suggestion. We have moved each research hypothesis on a new line to make them more visible in the text.
- Results
- Line 191: because it is the first appearance in the text, the full name for the SES and the abbreviation in parentheses must be used. After that, the abbreviation can be written in the text of the paper.
Response: Thank you very much for your comment. We have added the full name “socioeconomic status” before the abbreviation.
- Line 195: Table 1 - The abbreviations used in Table 1 are not indicated. It is recommended to include a table foot with the meaning of M and SD.
Response: Thank you very much for your comment. We have added notes of abbreviations in Table 1.
- Also, for Table 1 and Table 2 the authors should mention the source.
Response: Thank you very much for your suggestion. We have added notes in both tables. “Data are from the China Family Panel Studies (CFPS), Wave 1 (2010) and Wave 2 (2012) (https://www.isss.pku.edu.cn/cfps/index.htm)”

Reviewer 2 Report
Lines 70-79, Consider increasing the explanation of social capital.
Lines 80-104, The operational and conceptual definition of self efficacy, can be developed in more detail.
Consider the addition of a conceptual diagram.
Explain the Table 1.
I'd love to see a likelihood ratio test or a measure discussed explaining the goodness of fit for the best model.

Author Response
Response to Reviewer 2 Comments
- Lines 70-79, Consider increasing the explanation of social capital.
Response: Thank you for your suggestion. We have added more review on the social capital on pp.2.
- Lines 80-104, The operational and conceptual definition of self efficacy, can be developed in more detail.
Response: Thank you for your suggestion. We have added more review on the academic efficacy on pp.2-3.
- Consider the addition of a conceptual diagram.
Response: Thank you very much for your suggestion. We have added a conceptual framework of the current study on pp. 3.
- Explain the Table 1.
Response: Thank you very much for your comment. We have added more explanation of Table 1 on pp. 5. we have also added more notes to Table 1 for further clarification.
- I'd love to see a likelihood ratio test or a measure discussed explaining the goodness of fit for the best model.
Response: Thank you for your comment.
Two strategies were adopted to select the best fitting model. Both AIC difference, as well as deviance statistics, provided some evidence that the final model fitted the data well. We also added in the analytic approach part(pp.5) that “lower AIC values indicate a better-fit model”
Moreover, thank you so much for all the detailed comments and suggestions in the comment boxes of the text. We have revised the details following your suggestion.

Reviewer 3 Report
This is a well-written paper. The question is very important and well chosen. The hypotheses are very well derived from the literature used. The calculations are logical.
Two very minor comments:
Z 120:
which institute is meant? Is it located in China?
Table 1: Provinces: Others: is this Shanghai?
Author Response
Response to Reviewer 3 Comments
This is a well-written paper. The question is very important and well chosen. The hypotheses are very well derived from the literature used. The calculations are logical.
Response: Thank you very much for your comment!
Two very minor comments:
Z 120:
which institute is meant? Is it located in China?
Response: Thank you very much for your question. The Institute refers to “Institute of Social Science Survey (ISSS) at Peking University”. It is located in China. We have added the information in the text.
Table 1: Provinces: Others: is this Shanghai?
Response: Thank you so much for your question. As we briefly introduced in lines 128-132, in addition to the five main sampling provinces, the CFPS also sampled households from 20 additional provinces to comprise a national population-based sample, and these are what the “other” in Table 1 refers to. Following your comment, we have added notes in the table to clarify this.
We also added a footnote: The 20 provinces “include Jiangsu, Zhejiang, Fujian, Jiangxi, Anhui, Shandong, Hebei, Shanxi, Jilin, Heilongjiang, Guangxi Zhuang Autonomous Region, Hubei, Hunan, Sichuan, Guizhou, Yunnan, Tianjin, Beijing, Chongqing, Shaanxi.”
